# Bis-Iridoid Glycosides and Triterpenoids from *Kolkwitzia amabilis* and Their Potential as Inhibitors of ACC1 and ACL

**DOI:** 10.3390/molecules29245980

**Published:** 2024-12-18

**Authors:** Jiang Wan, Ze-Yu Zhao, Can Wang, Chun-Xiao Jiang, Ying-Peng Tong, Yi Zang, Yeun-Mun Choo, Jia Li, Jin-Feng Hu

**Affiliations:** 1Zhejiang Provincial Key Laboratory of Plant Evolutionary Ecology and Conservation, School of Pharmaceutical Sciences, Taizhou University, Taizhou 318000, China; jwan@tzc.edu.cn (J.W.); 21111030034@m.fudan.edu.cn (Z.-Y.Z.); wangcan@tzc.edu.cn (C.W.); jiangchx15@tzc.edu.cn (C.-X.J.); fish166@126.com (Y.-P.T.); 2Department of Natural Medicine, School of Pharmacy, Fudan University, Shanghai 201203, China; 3State Key Laboratory of Drug Research, Shanghai Institute of Materia Medica, Chinese Academy of Sciences, Shanghai 201203, China; yzang@simm.ac.cn (Y.Z.); jli@simm.ac.cn (J.L.); 4Chemistry Department, Faculty of Science, University of Malaya, Kuala Lumpur 50603, Malaysia; ymchoo@um.edu.my

**Keywords:** *Kolkwitzia amabilis*, bis-iridoid glycosides, triterpenoids, acetyl CoA carboxylase 1, ATP-citrate lyase

## Abstract

A comprehensive phytochemical investigation of the twigs/leaves and flower buds of *Kolkwitzia amabilis*, a rare deciduous shrub native to China, led to the isolation of 39 structurally diverse compounds. These compounds include 11 iridoid glycosides (**1**–**4** and **7**–**13**), 20 triterpenoids (**5**, **6**, and **14**–**31**), and 8 phenylpropanoids (**32**–**39**). Among these, amabiliosides A (**1**) and B (**2**) represent previously undescribed bis-iridoid glycosides, while amabiliosides C (**3**) and D (**4**) feature a unique bis-iridoid-monoterpenoid indole alkaloid scaffold with a tetrahydro-*β*-carboline-5-carboxylic acid moiety. Amabiliacids A (**5**) and B (**6**) are 24-nor-ursane-type triterpenoids characterized by an uncommon ∆^11,13(18)^ transannular double bond. Their chemical structures and absolute configurations were elucidated through spectroscopic data and electronic circular dichroism analyses. Compound **2** exhibited a moderate inhibitory effect against acetyl CoA carboxylase 1 (ACC1), with an IC_50_ value of 9.6 μM. Lonicejaposide C (**8**), 3*β*-*O*-*trans*-caffeoyl-olean-12-en-28-oic acid (**29**), and (23*E*)-coumaroylhederagenin (**31**) showed notable inhibitory effects on ATP-citrate lyase (ACL), with IC_50_ values of 3.6, 1.6, and 4.7 μM, respectively. Additionally, 3*β*-acetyl-ursolic acid (**17**) demonstrated dual inhibitory activity against both ACC1 and ACL, with IC_50_ values of 10.3 and 2.0 μM, respectively. The interactions of the active compounds with ACC1 and ACL enzymes were examined through molecular docking studies. From a chemotaxonomic perspective, the isolation of bis-iridoid glycosides in this study may aid in clarifying the taxonomic relationship between the genera *Kolkwitzia* and *Lonicera* within the Caprifoliaceae family. These findings highlight the importance of conserving plant species with unique and diverse secondary metabolites, which could serve as potential sources of new therapeutic agents for treating ACC1/ACL-associated diseases.

## 1. Introduction

The Caprifoliaceae family comprises 41 genera and approximately 960 species, primarily distributed in eastern North America and East Asian countries [1]. Several species within this family are cultivated not only as ornamental plants but also utilized in traditional Chinese medicine for treating various diseases [2,3]. Phytochemical investigations of Caprifoliaceae species have indicated that iridoids and triterpenoids are the predominant secondary metabolites [1,4,5]. Recent pharmacological studies have revealed that chemical constituents from Caprifoliaceae species exhibit a broad spectrum of bioactivities [1,4,5,6,7], particularly showing inhibitory effects against acetyl CoA carboxylase 1 (ACC1) and ATP-citrate lyase (ACL). For example, various structurally diverse iridoid and triterpenoid glycosides with ACC1 or ACL inhibitory activities have been isolated and characterized from the flower buds of two Caprifoliaceae species (*Lonicera japonica* and *Abelia* × *grandiflora*), as well as from the flower buds and twigs/leaves of *Heptacodium miconioides*, an endangered monotypic species in the Caprifoliaceae family [8,9].

*Kolkwitzia amabilis* Graebner, commonly known as ‘Beauty Brush’, is another monotypic species in the Caprifoliaceae family [10,11]. It earned its Chinese name, ‘Weishi’, due to its densely yellow bristled fruit, which resembles a hedgehog. This relict species is a deciduous shrub that grows 3–5 m tall, characterized by red-brown branches, ovate-elliptic leaves, and reddish corollas that bloom in early summer. Renowned for its outstanding striking spring flowers, *K. amabilis* displays prominently hairy calyces and stalks. This shrub has a dense, upright-arching, and vase-shaped, fountain-like growth habit. Nowadays, this species is cultivated as an ornamental plant [12]. *K. amabilis* is an exceedingly rare shrub native to the wilds of central China, where it was first collected in the mountains of Hubei province by plant explorer Earnest Henry Wilson in 1901. The small and sparse wild populations of *K. amabilis* hinder its natural regeneration. It is listed as a ‘rare’ species in the China Plant Red Data Book [13] and classified as a ‘third-grade’ nationally protected plant in China. Previous studies have conducted systematic and comprehensive investigations on its morphological phenotype traits and molecular phylogenetics [11,12]. However, phytochemical, pharmacological, and chemotaxonomic significance investigations of *K. amabilis* remain limited, with only its volatile organic compounds analyzed to date via GC-MS [14].

As part of an ongoing project aimed at discovering novel plant-derived agents for the treatment of ACC1/ACL-associated disease (e.g., hyperlipidemia and cardiovascular diseases) [8,9,15], a phytochemical investigation of *K. amabilis* cultivated at a botanical garden was conducted. This study led to the isolation of 4 new bis-iridoid glycosides (**1**–**4**) and 2 new 24-nor-ursane-type triterpenoids (**5** and **6**), along with 33 structurally diverse known compounds (**7**–**39**). Presented herein are the isolation, structure determination, inhibitory effects against ACC1 and ACL, and the chemotaxonomic significance of these compounds. This work constitutes Part XLII of the series titled “Phytochemical and biological studies on rare and endangered plants endemic to China” (for Parts XXXVIII through XLI [16,17,18,19]).

## 2. Results and Discussion

The present phytochemical investigations led to the isolation and characterization of 39 structurally diverse chemical constituents (**1**–**39**). Of these, 35 were obtained from the twigs/leaves, and 7 were isolated from the flower buds, with 3 compounds common to both sources. These constituents include 11 iridoid glycosides (**1**–**4** and **7**–**13**, Figure 1), 20 triterpenoids (**5**, **6**, and **14**–**31**, Figure 2), and 8 phenylpropanoids (**32**–**39**, Appendix A).

The 90% MeOH extract of *K. amabilis* twigs and leaves underwent sequential partitioning and chromatography, resulting in the isolation of compounds **1**–**10**, **12**–**34**, **37**, and **38**. By comparing observed spectroscopic data and physicochemical properties with reported values, the known compounds from the twigs and leaves were identified as saungmaygaoside C (**7**) [20], lonicejaposide C (**8**) [9], saungmaygaoside D (**9**) [20], kinginoside (**10**) [21], sweroside (**12**) [22], 8-*epi*-loganin (**13**) [23], 2,6*β*-dihydroxy-3-oxo-11*α*,12*α*-epoxy-24-norursa-1,4-dien-28,13*β*-olide (**14**) [24], 2*α*,3*β*-dihydroxy-lupa-12,20(29)-dien-28-oic acid (**15**) [25], ursolic acid (**16**) [26], 3*β*-acetyl-ursolic acid (**17**) [27], 3-oxo-ursolic acid (**18**), uvaol (**19**) [28], corosolic acid (**20) [29]**, 2*α*,23-dihydroxy-3*β*-(*trans*-p-coumaroyloxy)-urs-12-en-28-oic aid (**21**) [30], jacoumaric acid (**22**) [31], 3*β*-*O*-*trans*-ferulyl-2*α*-hydroxy-urs-12-en-28-oic acid (**23**) [32], oleanolic acid (**24**) [26], maslinic acid (**25**) [33], hederagenin (**26**) [34], erythrodiol (**27**) [35], 3*β*-*O*-*E*-coumaroylarjunolic acid (**28**) [36], 3*β*-*O*-*trans*-caffeoyl-olean-12-en-28-oic acid (**29**) [37], 3*β*-*O*-*trans*-caffeoyl-2*α*-hydroxy-olean-12-en-28-oic acid (**30**) [38], (23*E*)-coumaroylhederagenin (**31**) [39], (*E*)-isoferulaadehyde (**32**) [40], 3,4-dimethoxycinnamic acid (**33**) [41], 4-methoxycinnamic acid (**34**) [42], 4-*O*-demethyl rhaphidecursinol A (**37**) [43], and (−)-pinoresinol (**38**) [44].

Using similar isolation procedures on the 90% MeOH extract of *K. amabilis* flower buds, compounds **7**, **11**–**13**, **35**, **36**, and **39** were obtained (Figure 1 and Appendix A). By comparing observed spectroscopic data and physicochemical properties with reported values, the known compounds were identified as saungmaygaoside C (**7**) [20], desferuloylkinginoside (**11**) [21], sweroside (**12**) [22], 8-*epi*-loganin (**13**) [23], *trans*-ferulic acid (**35**) [45], coniferin (**36**) [46], and (7*S*,8*R*)-urolignoside (**39**) [47].

### 2.1. Structure Identification of Compounds ***1**–**6***

Amabilioside A (**1**) was isolated as a colorless oil, and the molecular formula was determined to be C_44_H_58_O_23_ based on the observed ion peak at *m*/*z* 953.3315 [M − H]^−^ (calcd. for C_44_H_57_O_23_ 953.3296) in HRESIMS and ^13^C NMR (Table 1) data. The ^1^H NMR (Table 1) spectrum of **1** displayed characteristic resonances for two oxygenated olefinic protons [*δ*_H_ 7.43 (br s, H-3) and 7.23 (br s, H-3′)], a set of terminal double bond [*δ*_H_ 5.74 (m, H-8), 5.26 (br d, *J* = 17.0 Hz, H-10a), and 5.20 (br d, *J* = 10.6 Hz, H-10b)], three acetal protons [*δ*_H_ 5.52 (d, *J* = 5.5 Hz, H-1), 5.47 (d, *J* = 1.8 Hz, H-1′), and 4.50 (dd, *J* = 6.9, 4.4 Hz, H-7)], one secondary methyl [*δ*_H_ 1.03 (d, *J* = 7.0 Hz, Me-10′)], and three methoxy group [*δ*_H_ 3.28, 3.28, and 3.15 (each 3H, s)]. Furthermore, the ^1^H NMR spectrum showed a pair of distinct *trans*-olefinic proton signals [*δ*_H_ 7.47 (d, *J* = 15.9 Hz, H-7″″) and 6.16 (d, *J* = 15.9 Hz, H-8″″)], as well as three ABX system aromatic proton signals [*δ*_H_ 7.03 (s, H-2″″), 6.94 (d, *J* = 8.0 Hz, H-6″″), and 6.78 (d, *J* = 8.0 Hz, H-5″″)]. These signals suggest the presence of an *E*-caffeoyl group in **1**, further confirmed by the absence of NOESY correlations between H-7″″ and H-8″″. Additionally, the characteristic signals of two anomeric protons [*δ*_H_ 4.86 (d, *J* = 8.1 Hz, H-1‴) and 4.69 (d, *J* = 7.9 Hz, H-1″)], along with 12 other highly overlapping protons ranging from *δ*_H_ 3.19 to 4.79, suggested the presence of two sugar moieties in **1**. Consistent with the above observations, the ^13^C NMR spectrum along with its HSQC data, displayed a total of 44 carbon signals, including three carbonyls (*δ*_C_ 168.7, 168.3, and 168.0), six aromatic carbons (*δ*_C_ 149.8, 147.0, 127.6, 123.2, 116.5, and 115.3), eight olefinic carbons (*δ*_C_ 153.3, 150.7, 147.0, 135.9, 119.8, 114.7, 114.5, and 112.0), three acetal carbons (*δ*_C_ 104.1, 97.8, and 95.5), and one oxygenated carbon.

(*δ*_C_ 78.6), and two glucoside units (*δ*_C_ 100.0, 97.6, 78.5, 78.4, 78.0, 75.8, 74.6, 74.5, 71.7, 71.5, 62.8, and 62.7). The above NMR data indicated that **1** is a bis-iridoid glycoside conjugated with an *E*-caffeoyl group. The ^1^H and ^13^C NMR spectroscopic data of the bis-iridoid glycoside moiety were similar to those of saungmaygaoside C (**7**) [20], a co-occurring bis-iridoid glycosides that was firstly isolated from *Picrorhiza kurroa*. The main difference is that the chemical shift at C-2‴ (*δ*_H_ 4.79 and *δ*_C_ 74.5 for **1**; *δ*_H_ 3.19 and *δ*_C_ 74.6 for **7**) in the glycosyl unit is significantly downfield. This suggests that the glycosyl and caffeoyl units are connected through an ester linkage between C-2‴ and C-9″″, which was confirmed by the ^1^H-^1^H COSY correlation of H-1‴/H-2‴/H-3‴/H-4‴/H-5‴/H_2_-6‴ and the HMBC correlation from H-2‴ to C-9″″ (Figure 3). The D-configuration of the sugar moieties in compound **1**, isolated in limited quantities, was confirmed by comparing the NMR data with those of the co-occurring bis-iridoid glycosides (saungmaygaoside C, **7**) and by considering biogenetic factors. Acid hydrolysis of **7** yielded only a monosaccharide, which was identified as D-glucose based on HPLC and optical rotation analysis. The retention time (*t*_R_ = 8.2 min, Appendix A) and optical rotation value (${\left[\alpha \right]}_{D}^{22}$ +70.2) of the hydrolysis product matched those of authentic D-glucose. Consequently, based on biogenetic considerations, the glycoside moiety in compound **1** is hypothesized to be analogous to that in **7**, specifically D-glucose. The coupling constant of 7.9 and 8.1 Hz between H-1″/H-1‴ and H-2″/H-2‴ indicated a diaxial relationship between these protons and a *β*-configuration at the anomeric position. The relative configuration of **1** was determined by analyzing the NOESY spectrum. The NOESY correlations (Figure 3) between H-8 with H-1/H_2_-6, H_3_-10′ with H-1′/H-7′, and H-5′/H-9′ with H-8′ indicated that the relative configuration of **1** was identical to that of **7**. Based on these findings, the structure of **1** was established as shown in Figure 1.

Amabilioside B (**2**) shares the same molecular formula (C_44_H_58_O_23_) as **1**, as determined from the HRESIMS and ^13^C NMR (Table 1) data. The primary differences are observed in the vicinity of C-2‴ and C-3‴, suggesting that the C-2‴ *E*-caffeoyl substituent in **1** is located at C-3‴ in **2**. This inference was confirmed by the HMBC (Appendix A) correlation from H-3‴ (*δ*_H_ 5.06, dd, *J* = 9.3, 9.2 Hz) to the carbonyl carbon (*δ*_C_ 169.0) in the *E*-caffeoyl moiety. The relative configuration of compound **2** was determined to be the same as that of **1** based on coupling constant (*J*) analysis and NOESY (Appendix A) data. Therefore, the structure of **2** was established as shown in Figure 1.

Amabilioside C (**3**) was obtained as a colorless oil. Its molecular formula was determined to be C_44_H_56_N_2_O_20_ based on the HRESIMS ion peak at *m*/*z* 933.3505 [M + H]^+^ (calcd. for C_44_H_57_N_2_O_20_ 933.3499) and ^13^C NMR data (Table 2), indicating 18 degrees of unsaturation. The IR spectrum showed the presence of hydroxyl (3416 cm^−1^) and carbonyl (1690 cm^−1^) functional groups. Interpretation of the ^1^H (Table 2) and ^13^C NMR data suggested that compound **3** possesses a typical bis-iridoid glycoside fragment, similar to compounds **1** and **2**, conjugated with an aromatic moiety. In addition to the NMR signals corresponding to the bis-iridoid glycoside moiety, the remaining resonances were consistent with a tetrahydro-*β*-carboline unit, featuring an ortho disubstituted phenyl group [*δ*_H_ 7.48 (d, *J* = 7.4 Hz, H-9); 7.04 (dd, *J* = 7.4, 7.0 Hz, H-10); 7.13 (dd, *J* = 8.0, 7.0 Hz, H-11); 7.31 (d, *J* = 8.0, H-12)] and a methine-amine [*δ*_H_ 3.89 (m, H-5)]. The NMR data of **3** closely resembled those of dipsaperine, which was previously isolated from the roots of *Dipsacus asper* [48]. Similar to dipsaperine, compound **3** is a hybrid of bis-iridoid glycoside and tetrahydro-*β*-carboline-5-carboxylic acid. This was confirmed by the ^1^H-^1^H COSY (Figure 4) correlations between H-5/H_2_-6 and the HMBC (Figure 4) cross-peaks from H-3 to C-2/C-5/C-7 and H-5 to C-3. The relative configuration of **3** was determined by analyzing the coupling constants and NOESY (Figure 4) experiments. The strong key NOESY cross-peaks between H_2_-14/H-19/H-21 revealed that the CH_2_-14 group, the terminal double bond, and H-21 were oriented on the same side and were assigned as *α*-orientation, thereby placing H-15/H-20 in the *β*-orientation. The NOESY correlation between H-3/H-5, along with the coupling constant of H-5 (*J* = 11.7 and 4.0 Hz), corresponds to those reported for dipsaperine [48] and 5(*S*)-5-carboxystrictosidine [49], thereby confirming the *α*-orientation of H-3 and H-5. Additionally, the secondary methyl group at C-8′ was found to be identical to those in compounds **1** and **2**, as confirmed by the NOESY experiment. The stereochemistry of C-3 and C-5 was assigned as *S*, based on the positive Cotton effect (CE) at 222 nm and negative CE at 236 nm observed in its ECD spectrum (Appendix A) [48], which were consistent with those of dipsaperine. Thus, the structure of **3** was determined as shown in Figure 1.

Amabilioside D (**4**) was found to possess the same molecular formula (C_44_H_56_N_2_O_20_) as compound **3**, as determined by the HRESIMS ion at *m*/*z* 933.3503 [M + H]^+^ (calcd. for C_44_H_57_N_2_O_20_ 933.3499) and ^13^C NMR data (Table 2). Detailed analysis of the 1D and 2D NMR (Appendix A) spectroscopic data revealed that **4** shares a similar structure as **3**, with a notable difference observed in the tetrahydro-*β*-carboline-5-carboxylic acid moiety. The two downfield chemical shifts of the methine-amine group in compound **4** [*δ*_H_ 4.84 (dd, *J* = 3.7, 3.7 Hz, H-3), 4.12 (dd, *J* = 9.1, 5.7 Hz, H-5); *δ*_C_ 52.7 (C-3), 55.0 (C-5)], were significantly distinct when compared to those in **3** [*δ*_H_ 4.51 (br d, *J* = 11.3 Hz, H-3), 3.89 (m, H-5); *δ*_C_ 53.2 (C-3), 59.6 (C-5)]. This suggests that the configuration at C-3 in **4** is inverted [50]. This was further supported by the absence of a NOESY correlation (Appendix A) between H-3 and H-5. Similar to compounds **1**–**3**, the Me-10′ in **4** was assigned an *α*-orientation, as indicated by the clear NOE cross-peaks between H_3_-10′ and H-1′/H-7′. Based on these data, the structure of **4** was determined as shown in Figure 1.

Amabiliacid A (**5**) was obtained as a white powder. Its molecular formula was determined as C_29_H_42_O_4_ based on HRESIMS (*m*/*z* 455.3157 [M + H]^+^, calcd. for 455.3156) and ^13^C NMR data (Table 3). In the upfield region of the ^1^H NMR spectrum of **5**, three tertiary methyl groups were observed at *δ*_H_ 0.75 (3H, s, Me-25), 0.90 (3H, s, Me-26), and 1.00 (3H, s, Me-27), along with two secondary methyl groups at *δ*_H_ 0.92 (3H, d, *J* = 7.1 Hz, Me-30) and 1.05 (3H, d, *J* = 7.2 Hz, Me-29). Additionally, signals at *δ*_H_ 5.16 and 4.73 (each 1H, br s) were observed for an exocylic double bond, along with two olefinic protons at *δ*_H_ 6.56 (1H, dd, *J* = 11.8, 2.9 Hz, H-11) and 5.72 (1H, d, *J* = 11.8 Hz, H-11), as well as two oxymethines at *δ*_H_ 3.76 (1H, d, *J* = 9.1 Hz, H-3) and 3.54 (1H, m, H-2). The ^13^C NMR spectrum displayed a total of 29 carbon signals, including a carboxyl carbon at *δ*_C_ 180.3 (C-28), six olefinic carbons at *δ*_C_ 105.0 (C-23), 126.5 (C-11), 127.6 (C-12), 136.8 (C-18), 138.9 (C-13), and 151.5 (C-4), as well as two oxygenated carbons at *δ*_C_ 74.2 (C-2) and 79.6 (C-3). The NMR data of **5** closely resembled those of corosolic acid (**20**) [29], a co-occurring ursane-type triterpenoid. A notable difference between them was the absence of the Me-23 and Me-24 groups in the NMR spectra of **5**. Instead, signals typical of two olefinic protons [*δ*_H_ 5.16 and 4.73 (br s)] corresponding to an exocyclic double bond functionality were observed. This suggests that **5** possesses a 24-nor-ursane-type triterpenoid framework. This deduction was confirmed by the ^1^H-^1^H COSY (Appendix A) correlations of H-3/H_2_-23 and H-5/H_2_-23, as well as the HMBC (Appendix A) correlation from H_2_-23 to C-3/C-4/C-5. Additionally, the olefinic group in **5** shifted to form an ∆^11,13(18)^ diene, as opposed to the ∆^12^ configuration in **20**. This was further supported by the ^1^H-^1^H COSY correlations of H-9/H-11/H-12 and key HMBC cross-peaks from H-9 to C-11/C-12, H-11 to C-13, H-12 to C-13/C-18, H_3_-27 to C-13, and H_3_-29 to C-18. The relative configuration of **5** was determined by analyzing the coupling constants (*J*) and NOESY data. The large *J* value between H-2 and H-3 (9.1 Hz), along with the NOE (Appendix A) correlations of H-2/H_3_-25 and H-3/H-5, confirmed the *α*-orientation of OH-2 and *β*-orientation of OH-3, respectively. Additionally, the ECD data (Appendix A) of **5** closely matched those of 2*α*,3*β*-dihydroxy-24-nor-olean-4(23),11,13(18)-trien-28-oic acid [51] and pterohoonoid D [52], two pentacyclic triterpenoids featuring a unique ∆^11,13(18)^ diene group in rings C and D. This comparison allowed the assignment of the 9*R*, 14*S*, and 17*S* configuration in **5**. Consequently, the structure of **5** was defined as (2*R*,3*R*,5*R*,8*R*,9*R*,10*S*,14*S*,17*S*,19*S*,20*R*)-2*α*,3*β*-dihydroxy-24-nor-urs-4(23),11,13(18)-trien-28-oic acid.

The molecular formula of amabiliacid B (**6**) was determined to be C_29_H_40_O_4_ based on HRESIMS analysis (*m*/*z* 453.2996 [M + H]^+^, calcd. for 453.2999) and ^13^C NMR data (Table 3), indicating two fewer protons than compound **5**. The NMR spectroscopic data of **6** were similar to those of **5**, except for the presence of the ∆^20(21)^ group [*δ*_H_ 1.66 (3H, s, H-30), 5.43 (1H, br d, *J* = 5.0 Hz, H-21); *δ*_C_ 137.6 (C-20), 121.0 (C-21)]. This was further corroborated by the HMBC (Appendix A) cross-peaks, which showed correlations between H_3_-20 with C-20 and H_3_-30 with C-20/C-21. The relative configuration of **6** was found to be identical to that of **5**, as evidenced by the NOESY (Appendix A) experiment. Consequently, the structure of **6** was determined to be 2*α*,3*β*-dihydroxy-24-nor-urs-4(23),11,13(18),20-tetraen-28-oic acid.

### 2.2. Anti-ACC1 and Anti-acl Bioactivities of the Isolated Compounds

ACC1 is a key enzyme in the de novo fatty acid biosynthesis pathway, catalyzing the conversion of acetyl-CoA to malonyl-CoA [53]. In contrast, ACL plays a critical role in linking carbohydrate and lipid metabolism by producing acetyl-CoA from citrate, which is essential for the biosynthesis of fatty acids and cholesterol [54]. Both enzymes are considered potential therapeutic targets for hyperlipidemia and other metabolic disorders. Firsocostat (formerly named ND-630 or GS-0976) is a liver-targeted allosteric ACC1 inhibitor currently in clinical stages, and it has demonstrated efficacy in reducing hepatic fat content in patients with non-alcoholic steatohepatitis (NASH) [55]. Bempedoic acid, a synthetic dicarboxylic acid-type ACL inhibitor, was approved by FDA in 2020 for the treatment of statin-resistant hypercholesterolemia [56]. As part of our ongoing efforts to discover novel plant-derived compounds for the treatment of ACC1/ACL-related diseases, we evaluated all isolated compounds (**1**–**39**) for their ACC1 and ACL inhibitory activities. Among these, compounds **2** (IC_50_: 9.6 μM) and **17** (IC_50_: 10.3 μM) displayed moderate inhibitory effects against ACC1. Compounds **8**, **17**, **29**, and **31** exhibited potent inhibitory effects on ACL, with IC_50_ values of 3.6, 2.0, 1.6, and 4.7 μM (Table 4 and Figure 5), respectively. The rest of the isolated compounds are inactive (inhibition percentage < 50% at 20 μM). The known inhibitor ND-630 (IC_50_: 2.0 ± 0.1 nM) [57] and BMS 303141 (IC_50_: 0.3 ± 0.1 μM) [58] were used as positive controls for ACC1 and ACL, respectively.

### 2.3. Molecular Docking Simulation of Compounds ***2***, ***17***, and ***29***

Considering the observed ACC1 and ACL inhibitory activities (Table 4), the dual inhibitory compound (**17** for ACC1 and ACL) and the most potent individual inhibitor (**2** for ACC1; **29** for ACL), were selected for molecular docking studies. These studies aimed to elucidate their potential mechanisms of action as inhibitors of ACC1 (PDB ID: 3TVU) and ACL (PDB ID: 6HXH). The structures of compounds **2**, **17**, and **29** exhibit favorable binding within the active site pockets of the selected target protein. The binding energies of compound **17** with ACC1 and ACL are −8.8 and −10.4 kcal/mol, respectively. In compound **17**, the carbonyl group establishes two hydrogen-bond interactions with Ser138 (3.0 and 2.8 Å) for ACC1 (Appendix A), and forms hydrogen bonds with Leu266 (2.8 Å) and Thr267 (2.9 Å) for ACL (Appendix A). The binding mode of compound **2** with ACC1 is shown in Figure 6, demonstrating a binding affinity of −8.8 kcal/mol. The hydroxyl groups in **2** formed four hydrogen bonds with Ile1820 (3.5 Å), Glu2177 (4.0 Å), Gly1579 (2.8 Å), and Arg1578 (3.6 Å) (Figure 6). Additionally, the phenyl ring in **2** formed a π-π stacking interaction with Try1576 (5.2 Å). Compound **29** exhibited a binding energy of −8.8 kcal/mol with ACL, with its hydroxyl and carboxyl groups positioned within the hydrophilic region of the binding site, forming four strong hydrogen-bond interactions with Glu599 (3.2 Å), Gly261 (3.3 Å), Gln505 (3.0, 4.1 Å), and Phe572 (3.0 Å) (Appendix A). Additionally, the carboxylate and carboxyl groups of **29** formed two salt bridge interactions with Lys1017 (4.7 Å) and Lys1018 (5.5 Å). Compounds **2**, **17**, and **29** exhibited multiple hydrogen bond interactions within the active binding sites of ACC1 and ACL. These interactions are fundamental to protein–ligand binding and likely contribute to the observed inhibitory activity of these compounds.

### 2.4. Chemotaxonomic Significance

*K. amabilis* remains rare in the wild but is widely cultivated in China, USA, and several European countries. As a member of the Caprifoliaceae family (commonly known as the honeysuckle family), this shrub represents a monotypic species and is renowned for its abundant blossoms, which display their full floral elegance in late spring or early summer. Comparative studies have been conducted on the morphology, growth, and flowering responses of various *Kolkwitzia* accessions grown under the same environmental conditions. Characterization of *K. amabilis* accessions based on flowering patterns and molecular markers was first discussed in 2014 [11]. Its complete chloroplast genome was sequenced, and phylogenetic analysis using a maximum parsimony approach indicated a close relationship between *K. amabilis* and *Lonicera japonica* [10], a species in the honeysuckle family. The ability to delimit species accurately is crucial for assessing biodiversity, a necessary step in establishing effective conservation policies [59]. Plant taxonomy has traditionally relied on morphological phenotype traits, it has progressively incorporated additional advanced techniques, including molecular phylogenetic and chemotaxonomy analyses [60,61,62,63,64,65]. The integrative taxonomy approach (ITA), which combines morphological and molecular phylogenetic evidence, has been recognized as one of the most effective methods for resolving complex species boundaries [66,67,68,69]. The bis-iridoids (**1**–**4** and **7**–**9**) isolated herein from the twigs/leave and flower buds of *K. amabilis* exhibit a secoiridoid/iridoid heterodimer structure, composed of secologanic acid derivatives linked to the 7-OH group of loganin. Most secoiridoid/iridoid-subtype dimers have previously been identified within the Caprifoliaceae family [70,71,72,73,74], indicating their potential as valuable chemotaxonomic markers. From a chemotaxonomic perspective, these findings further support the close phylogenetic relationship between *Kolkwitzia* and *Lonicera*.

## 3. Materials and Methods

### 3.1. General Experimental Procedures

Optical rotations were recorded in MeOH with an SGW-1 digital polarimeter (Shanghai INESA Physico-Optical Instrument Co., Ltd., Shanghai, China). Ultraviolet spectra were measured with a Shimadzu PharmaSpec-1800 UV-visible spectrometer (Shimadzu, Kyoto, Japan). IR spectra were obtained on a Thermo Nicolet iS10 spectrophotometer (Thermo Fisher Scientific, San Jose, CA, USA) in KBr discs. The 1D and 2D NMR spectra were collected on Bruker Avance III 400 MHz NMR instruments (Bruker BioSpin, Rheinstetten, Germany). Chemical shifts were reported in parts per million with the residual solvent signals as an internal standard. HRESIMS data were acquired on a Thermo Fisher Q-Exactive mass spectrometer (Thermo Fisher Scientific) with heated electrospray ionization. Semi-preparative HPLC was performed on a Shimadzu CBMA-20A system equipped (Shimadzu) with an SPD-20A photodiode array detector (Shimadzu) using the Cosmosil Packed Column (10 × 250 mm, 5 μm, 5C_18_-MS-II or π NAP) with a flow rate of 3 mL/min. Silica gel (Qing Dao Hai Yang Chemical Group Co., Qingdao, China; 200–300 mesh), MCI gel CHP20P (75–150 μm, Mitsubishi Chemical Industries, Tokyo, Japan), and Sephadex LH-20 gel (Amersham Biosciences, Slough, UK; 40–70 μm) were used for column chromatography (CC). Thin-layer chromatography (TLC) was performed on percolated plates (GF_254_, 0.2–0.25 mm, Qing Dao Hai Yang Chemical Group Co., Qingdao, China). TLC spots were visualized under UV light (254 or 365 nm) and by spaying with 5% H_2_SO_4_/vanillin followed by heating to 120 °C.

### 3.2. Plant Materials

The twigs/leaves of *Kolkwitzia amabilis* were harvested from Jiashan Lige Ecological Technology Co., Ltd., Jiaxing, Zhejiang province, China, in November 2021, and the flower buds were collected from the same source in April 2023. These plant samples were taxonomically identified by Prof. Ze-Xin Jin (Taizhou University, Taizhou, China). Voucher specimens (twigs/leaves: No. 20211107; flower buds: No. 20230420) have been deposited in the Herbarium of the Institute of Natural Medicine and Health Products, School of Pharmaceutical Science, Taizhou University.

### 3.3. Extraction and Isolation

The air-dried and milled twigs and leaves of *K. amabilis* (12.0 kg) were percolated four times with 90% MeOH (20 L) at ambient temperature. The extract was concentrated under reduced pressure, yielding a residue (2.2 kg) that was suspended in H_2_O and sequentially partitioned with petroleum ether (PE, 3 × 4 L), EtOAc (3 × 4 L), and *n*-BuOH (3 × 4 L). Following solvent removal, the EtOAc-soluble extract (231 g) was subjected to column chromatography (CC) on a silica gel column and separated into six fractions (Fr.1–Fr.6) using a gradient system of PE/EtOAc (10:1, 5:1, 3:1, 1:1, and 0:1, *v*/*v*). Fr.2 was applied to a Sephadex LH-20 column, followed by semi-preparative HPLC (MeCN-H_2_O, 65:35), to yield compound **17** (8.0 mg, *t*_R_ = 12.6 min). Fr.3 was subjected to MCI column chromatography using a MeOH-H_2_O gradient (50:50, 70:30, 85:15, and 100:0, *v*/*v*), resulting in four fractions (Fr.3.1–Fr.3.4). Compounds **33** (2.3 mg, *t*_R_ = 14.3 min) and **34** (1.7 mg, *t*_R_ = 13.4 min) were obtained from Fr.3.2 via semi-preparative HPLC (MeOH-H_2_O, 40:60). Fr.3.3 was further fractionated using Sephadex LH-20 column chromatography (MeOH), yielding five subfractions (Fr.3.3.1–Fr.3.3.5) based thin-layer chromatography (TLC) spots analysis. Compounds **15** (0.8 mg, *t*_R_ = 11.7 min) and **27** (7.4 mg, *t*_R_ = 13.9 min) were obtained from Fr.3.3.2 via semi-preparative HPLC (MeCN-H_2_O, 75:25), while compounds **19** (13.5 mg, *t*_R_ = 14.7 min) and **18** (2.7 mg, *t*_R_ = 17.1 min) were isolated from Fr.3.3.4 using semi-preparative HPLC (MeCN-H_2_O, 70:30). Fr.4 was separated by gel permeation chromatography on Sephadex LH-20 (MeOH), followed by semi-preparative HPLC using MeCN-H_2_O (75:25) to yield compounds **24** (2.4 mg, *t*_R_ = 11.6 min) and **16** (14.2 mg, *t*_R_ = 12.1 min). Fr.5 was subjected to MCI column chromatography with MeOH-H_2_O gradient (50:50, 70:30, 85:15, and 100:0, *v*/*v*), yielding seven subfractions (Fr.5.1–Fr.5.7). Purification of Fr.5.2 via semi-preparative HPLC (MeCN-H_2_O, 30:70) provided compound **32** (1.5 mg, *t*_R_ = 15.0 min). Further separation of Fr.5.3 over Sephadex LH-20 with MeOH, followed by semi-preparative HPLC (MeOH-H_2_O, 70:30), afforded compounds **37** (4.2 mg, *t*_R_ = 18.3 min) and **38** (6.1 mg, *t*_R_ = 11.6 min). Fr.5.4 was further fractionated using Sephadex LH-20 with MeOH, yielding four fractions (Fr.5.4.1–Fr.5.4.4). Separation of Fr.5.4.1 by semi-preparative HPLC (MeOH-H_2_O, 90:10) afforded compounds **26** (2.9 mg, *t*_R_ = 12.0 min) and **25** (6.1 mg, *t*_R_ = 13.1 min). Subsequently, compounds **5** (3.0 mg, *t*_R_ = 9.2 min) and **20** (1.0 mg, *t*_R_ = 15.3 min) were purified from Fr.5.4.2 by semi-preparative HPLC with an isocratic elution (MeCN-H_2_O, 65:35). Compounds **6** (1.0 mg, *t*_R_ = 14.8 min), **14** (4.1 mg, *t*_R_ = 11.0 min), **28** (3.5 mg, *t*_R_ = 15.0 min), and **21** (2.1 mg, *t*_R_ = 15.9 min) were obtained from Fr.5.4.3 via semi-preparative HPLC (MeOH-H_2_O, 85:15), respectively. Fr.5.4.4 was purified by semi-preparative HPLC (MeCN-H_2_O, 60:40) to furnish compound **29** (2.1 mg, *t*_R_ = 13.6 min). Fr.5.5 was fractionated using Sephadex LH-20 (MeOH), producing subfractions Fr.5.5.1–Fr.5.5.5. Compounds **31** (0.9 mg, *t*_R_ = 9.6 min), **30** (1.0 mg, *t*_R_ = 12.5 min), **22** (3.9 mg, *t*_R_ = 13.6 min), and **23** (2.9 mg, *t*_R_ = 15.0 min) were isolated from Fr.5.5.4 by semi-preparative HPLC (MeCN-H_2_O, 70:30).

Using D-101 macroporous resin column chromatography with a stepwise gradient-elution of MeOH-H_2_O (30:70, 50:50, 70:30, 90:10, and 100:0, *v*/*v*), the *n*-BuOH-soluble fraction (344 g) was divided into seven fractions (Fr.7–Fr.13). Purification of Fr.9 using Sephadex LH-20 (MeOH), followed by semi-preparative HPLC (MeCN-H_2_O, 15:85), gave compounds **12** (15.7 mg, *t*_R_ = 13.4 min) and **13** (10.4 mg, *t*_R_ = 16.2 min). Fr.11 was further separated using ODS C-18 CC (MeOH-H_2_O, 50:50, 60:40, 70:30, and 100:0, *v*/*v*), yielding eight subfractions (Fr.11.1–Fr.11.8). Fr.11.3 was purified by semi-preparative HPLC (MeCN-H_2_O, 30:70) to furnish compound **7** (3.2 mg, *t*_R_ = 15.9 min). Compounds **8** (8.7 mg, *t*_R_ = 11.9 min), **9** (8.7 mg, *t*_R_ = 9.4 min), and **4** (5.6 mg, *t*_R_ = 15.8 min) were obtained from Fr.11.4 by semi-preparative HPLC (MeOH-H_2_O, 55:45). Further purification of Fr.11.6 by Sephadex LH-20 (MeOH), followed by semi-preparative HPLC (MeCN-H_2_O, 30:70) gave compounds **10** (14.0 mg, *t*_R_ = 9.7 min) and **3** (4.5 mg, *t*_R_ = 14.1 min). Compounds **2** (2.6 mg, *t*_R_ = 15.1 min) and **1** (4.6 mg, *t*_R_ = 18.3 min) were obtained from Fr.11.7 using Sephadex LH-20 (MeOH), followed by semi-preparative HPLC (MeCN-H_2_O, 35:65).

The powdered flower buds (300 g) of *K. amabilis* were extracted with 90% methanol (4 × 2 L, each extraction for 24 h) at room temperature. The resulting dark green residue (41.0 g) was suspended in H_2_O (1.0 L) and successively extracted with petroleum ether (PE, 3 × 1.0 L), EtOAc (3 × 1.0 L), and *n*-BuOH (3 × 1.0 L). The EtOAc-soluble extract (36.3 g) was subjected to silica gel column chromatography and eluted with a PE/EtOAc gradient system (10:1, 5:1, 3:1, 1:1, and 0:1, *v*/*v*), yielding four fractions (Fr.14–Fr.17). Purification of Fr.15 using Sephadex LH-20 (MeOH), followed by semi-preparative HPLC (MeOH-H_2_O, 50:50) afforded compound **35** (2.7 mg, *t*_R_ = 12.8 min). The *n*-BuOH-soluble extract (19.3 g) was fractioned on an MCI column with a stepwise gradient elution of MeOH-H_2_O (30:70, 50:50, 70:30, 85:15, 100:0, *v*/*v*), yielding six fractions (Fr.18–Fr.23). Compound **36** (1.4 mg, *t*_R_ = 11.3 min) was obtained from Fr.19 by semi-preparative HPLC (MeOH-H_2_O, 25:75). Fr.20 was purified using Sephadex LH-20 (MeOH), followed by semi-preparative HPLC (MeOH-H_2_O, 30:70) to furnish compounds **12** (11.5 mg, *t*_R_ = 15.7 min) and **13** (3.5 mg, *t*_R_ = 16.5 min). Compound **39** (2.6 mg, *t*_R_ = 17.7 min) was obtained from Fr.21 using Sephadex LH-20 (MeOH), followed by semi-preparative HPLC (MeOH-H_2_O, 35:65). Fr.23 was further fractionated using Sephadex LH-20 (MeOH), yielding four subfractions (Fr.23.1–Fr.23.5). Separation of Fr.23.3 was performed by semi-preparative HPLC (MeOH-H_2_O, 50:50), resulting in compound **7** (30.1 mg, *t*_R_ = 10.8 min), while compound **11** (4.7 mg, *t*_R_ = 10.1 min) was purified from Fr.23.4 by semi-preparative HPLC with an isocratic elution (MeOH-H_2_O, 40:60).

Amabilioside A (**1**): colorless oil; ${\left[\alpha \right]}_{D}^{22}$ −57 (*c* 0.1, MeOH); UV (MeOH) λ_max_ (log ε) 226 (1.72), 328 (4.31) nm; IR (KBr) ν_max_ 3460, 1687, 1637, 1555, 1465, 1382, 1310, 1255, 1125, 1082, 1017, and 627 cm^−1^; ^1^H and ^13^C NMR data, see Table 1; HRESIMS *m*/*z* 953.3315 [M − H]^−^ (calcd. for C_44_H_57_O_23_ 953.3296, ∆ = 2.0 ppm).

Amabilioside B (**2**): colorless oil; ${\left[\alpha \right]}_{D}^{22}$ −68 (*c* 0.1, MeOH); UV (MeOH) λ_max_ (log ε) 227 (1.36), 324 (3.71) nm; IR (KBr) ν_max_ 3455, 1628, 1513, 1445, 1383, 1310, 1250, 1118, 1084, 1012, 747, and 635 cm^−1^; ^1^H and ^13^C NMR data, see Table 1; HRESIMS *m*/*z* 953.3306 [M − H] ^−^ (calcd. for C_44_H_57_O_23_ 953.3296, ∆ = 1.0 ppm).

Amabilioside C (**3**): colorless oil; ${\left[\alpha \right]}_{D}^{22}$ −113 (*c* 0.1, MeOH); UV (MeOH) λ_max_ (log ε) 224 (3.48); ECD (*c* 2.11 × 10^−3^ M, MeOH) λ_max_ (log ε) 206 (−7.7), 222 (+9.2), 236 (−31.3) nm; IR (KBr) ν_max_ 3497, 1637, 1557, 1445, 1385, 1310, 1207, 1110, 1080, 987, 747, and 620 cm^−1^; ^1^H and ^13^C NMR data, see Table 2; HRESIMS *m*/*z* 933.3505 [M + H]^+^ (calcd. for C_44_H_57_N_2_O_20_ 933.3499, ∆ = 0.6 ppm).

Amabilioside D (**4**): colorless oil; ${\left[\alpha \right]}_{D}^{22}$ −87 (*c* 0.1, MeOH); UV (MeOH) λ_max_ (log ε) 225 (2.74) nm; ECD (*c* 1.34 × 10^−3^ M, MeOH) λ_max_ (log ε) 234 (−18.6), 268 (−11.0) nm; IR (KBr) ν_max_ 3456, 1641, 1555, 1445, 1380, 1306, 1214, 1108, 1082, 989, and 630 cm^−1^; ^1^H and ^13^C NMR data, see Table 2; HRESIMS *m*/*z* 933.3503 [M + H]^+^ (calcd. for C_44_H_57_N_2_O_20_ 933.3499, ∆ = 0.4 ppm).

Amabiliacid A (**5**): white powder; ${\left[\alpha \right]}_{D}^{22}$ −67 (*c* 0.1, MeOH); UV (MeOH) λ_max_ (log ε) 250 (3.77) nm; ECD (*c* 1.52 × 10^−3^ M, MeOH) λ_max_ (log ε) 201 (+23.7), 223 (+9.2), 252 (−32.4) nm; IR (KBr) ν_max_ 3454, 1680, 1637, 1562, 1455, 1382, 1117, 1015, 747, and 620 cm^−1^; ^1^H and ^13^C NMR data, see Table 3; HRESIMS *m*/*z* 455.3156 [M + H]^+^ (calcd. for C_29_H_43_O_4_ 455.3156, ∆ = 0 ppm).

Amabiliacid B (**6**): white powder; ${\left[\alpha \right]}_{D}^{22}$ −46 (*c* 0.1, MeOH); UV (MeOH) λ_max_ (log ε) 248 (2.31) nm; IR (KBr) ν_max_ 3417, 1680, 1442, 1361, 1128, 1015, and 620 cm^−1^; ^1^H and ^13^C NMR data, see Table 3; HRESIMS *m*/*z* 453.2995 [M + H]^+^ (calcd. for C_29_H_41_O_4_ 453.2996, ∆ = −0.2 ppm).

### 3.4. Acid Hydrolysis of Saungmaygaoside C (***7***)

Compound **7** (4.0 mg) was dissolved in 2M aqueous CF_3_COOH (10 mL), and the solution was heated at 90 °C for 6 h, then allowed to cool to room temperature. After extracting with CHCl_3_, the aqueous layer was evaporated, yielding a monosaccharide extract (2 mg). The identification of the monosaccharide extract was conducted on an Agilent 1260 Infinity II HPLC system (Agilent, Santa Clara, CA, USA) coupled with an Evaporative Light Scattering Detector (ELSD) (Agilent, Santa Clara, CA, USA) [TSKgel Amide-80 5 μm column (4.6 × 250 mm); mobile phase: MeCN-H_2_O, 70:30; detection: ELSD; flow rate: 1.0 mL/min]. The peak at *t*_R_ = 8.2 min matched that of D-glucose [D-(+)-glucose, Batch # 088K00391, Sigma-Aldrich, St. Louis, MO, USA]. The D-configuration was confirmed by its optical rotation value of ${\left[\alpha \right]}_{D}^{22}$ +70.2 (*c* 0.2, H_2_O) {lit [75]: ${\left[\alpha \right]}_{D}^{23}$ +44.0 (*c* 0.1, H_2_O)}.

### 3.5. ACC1 and ACL Inhibitory Assay

The ACC1 and ACL inhibitory activity assay were conducted using ADP-Glo^TM^ luminescence assay reagents (Promega Biotech Co., Ltd, Beijing, China), which measure ACC1 or ACL activity by quantifying the amount of ADP generated in the enzymatic reaction [58]. The known inhibitor ND-630 (ACC1) [57] and BMS 303141 (ACL) [58] were used as positive controls, following previously described procedures [9,15]. The kinase assay was performed in a 384-well plate (ProxiPlateTM-384 Plus, PerkinElmer) with a 5 μL reaction mixture comprising 2.0 μL of ACC1 (or ACL), 2.0 μL of ATP, and 1.0 μL of the tested compounds at varying concentrations. Reactions were conducted in each well for 30 min at room temperature. Following this, 2.0 μL of ATP was added to each well, and the mixture was incubated for 1 h at room temperature. Negative control wells, which lacked the protein kinase, substrate, and ATP, were also included in the kinase assay. After the enzymatic reaction, 2.5 μL of ADP-Glo reagent was added to each well to terminate the kinase reaction and deplete any unconsumed ATP within 1 h at room temperature. Finally, 5.0 μL of kinase detection reagent was added to each well and incubated for 1 h to convert ADP to ATP. The luminescence signal was measured using a PerkinElmer EnVision reader.

### 3.6. Molecular Docking Simulation

Molecular docking simulations were primarily conducted using the AutoDock software package (http://autodock.scripps.edu/ accessed on 15 May 2024, The Scripps Research Institute, La Jolla, CA, USA). The three-dimensional (3D) structure of ACC1 (PDB ID: 3TVU) and ACL (PDB ID: 6HXH) [76] were retrieved from the RCSB Protein Data Bank (https://www.rcsb.org/ accessed on 15 May 2024). The 3D chemical structures of the ligands (compounds **2**, **17**, and **29**) were subjected to MM2 energy minimization using Chem3D (14.0) software (PerkinElmer, Waltham, MA, USA). AutoDock tools (http://mgltools.scripps.edu/ accessed on 15 May 2024) were used to prepare both ACC1/ACL and the ligands for docking. The protein preparation procedure involved removing water molecules and adding missing polar hydrogen atoms. The docking grid was centered on the ligand position, with the bounding box parameter set to 20 Å. The number of GA (Genetic Algorithm) runs was set to 20, and all other parameters were kept at their default settings. Binding free energy was calculated, and the pose with the lowest binding affinity to the protein was selected as the best conformation. Finally, the docking conformation was visualized using PyMOL (2.5.2) software (https://pymol.org/ accessed on 15 May 2024).

## 4. Conclusions

In conclusion, this study identifies the twigs/leaves and flower buds of *K. amabilis* as a rich source of iridoid glycosides (**1**–**4** and **7**–**13**), triterpenoids (**5**, **6**, and **14**–**31**), and phenylpropanoids (**32**–**39**), marking the first detailed report of its chemical constituents. Compounds **3** and **4** exhibit a bis-iridoid-monoterpenoid indole alkaloid scaffold featuring a tetrahydro-*β*-carboline-5-carboxylic acid moiety, while compounds **5** and **6** are 24-nor-ursane-type triterpenoids with a rare ∆^11,13(18)^ transannular double bond configuration. These findings greatly enrich the structural diversity of bis-iridoids and triterpenoids within the Caprifoliaceae family. Together with previous studies, the present results suggest that secoiridoid/iridoid-subtype dimers may serve as valuable chemical markers for this family [8,9,70,71,72,73,74]. Furthermore, several of these structurally intriguing bis-iridoid glycosides and triterpenoids (**2**, **8**, **17**, **29**, and **31**) demonstrate ACC1 or ACL inhibitory activity, indicating their potential as alternative therapeutic agents for treating metabolic diseases. These findings highlight the value of rare horticultural plants as significant natural resources in the discovery of novel bioactive compounds.

## Figures and Tables

**Figure 1 molecules-29-05980-f001:**
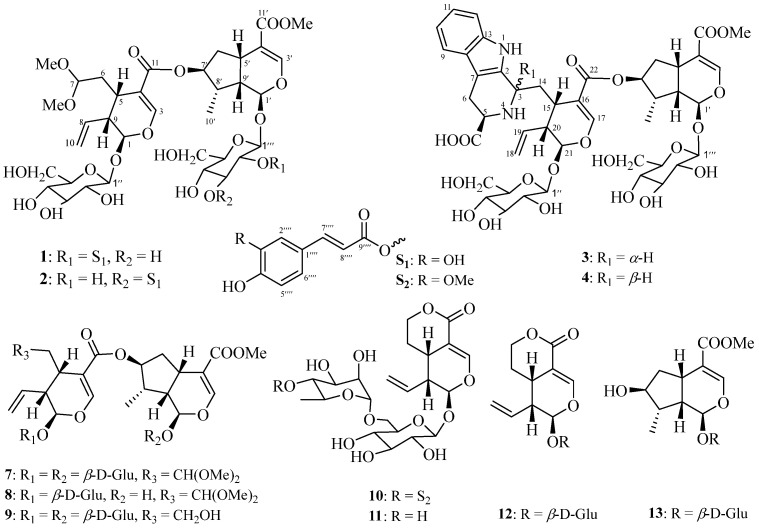
Iridoids (**1**–**4** and **7**–**13**) from the twigs/leaves and flower buds of *K. amabilis*.

**Figure 2 molecules-29-05980-f002:**
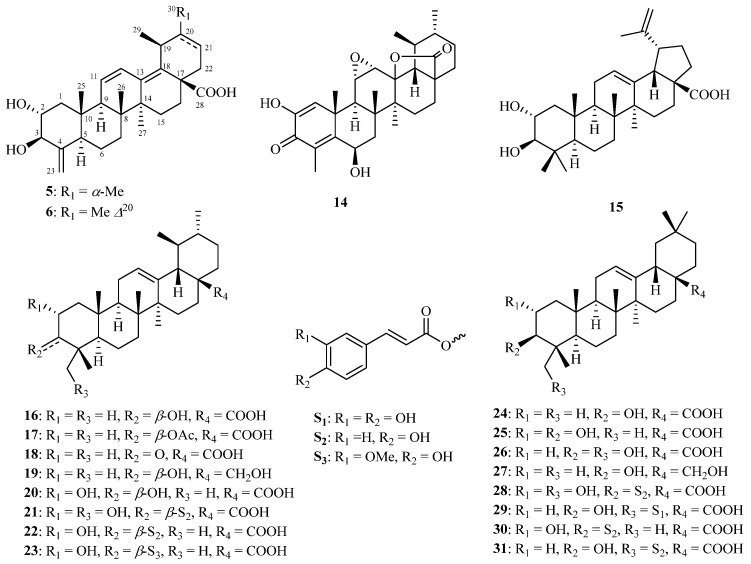
Triterpenoids (**5**, **6**, and **14**–**31**) from the twigs and leaves of *K. amabilis*.

**Figure 3 molecules-29-05980-f003:**
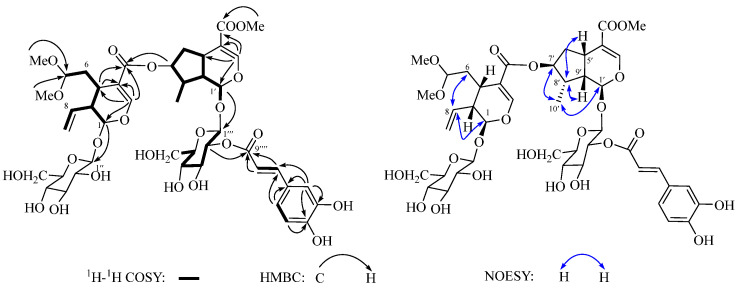
Key ^1^H-^1^H COSY, HMBC, and NOESY correlations of compound **1**.

**Figure 4 molecules-29-05980-f004:**
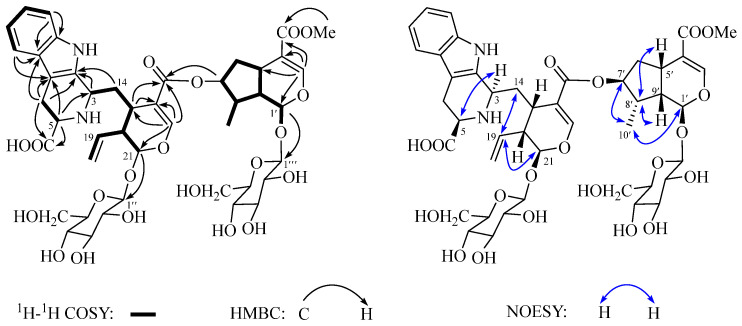
Key ^1^H-^1^H COSY, HMBC, and NOESY correlations of compound **3**.

**Figure 5 molecules-29-05980-f005:**
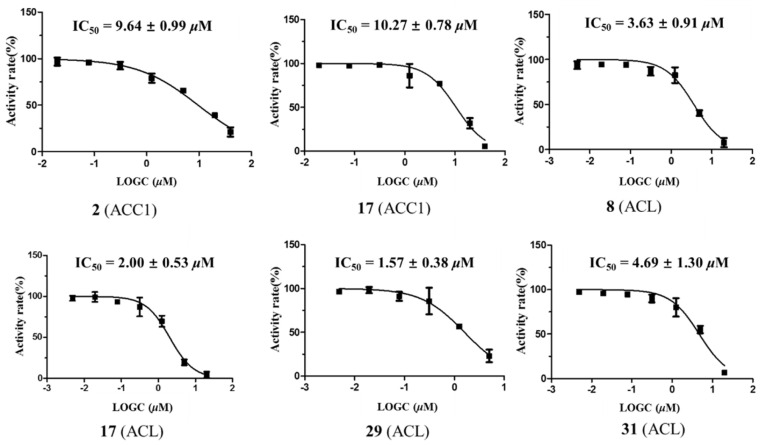
Acetyl CoA carboxylase 1 (ACC1) inhibitory IC_50_ values for compounds **2** and **17**; ATP-citrate lyase (ACL) inhibitory IC_50_ values for compounds **8**, **17**, **29**, and **31**. Each bar represents the mean ± standard deviation (SD) (*n* = 3).

**Figure 6 molecules-29-05980-f006:**
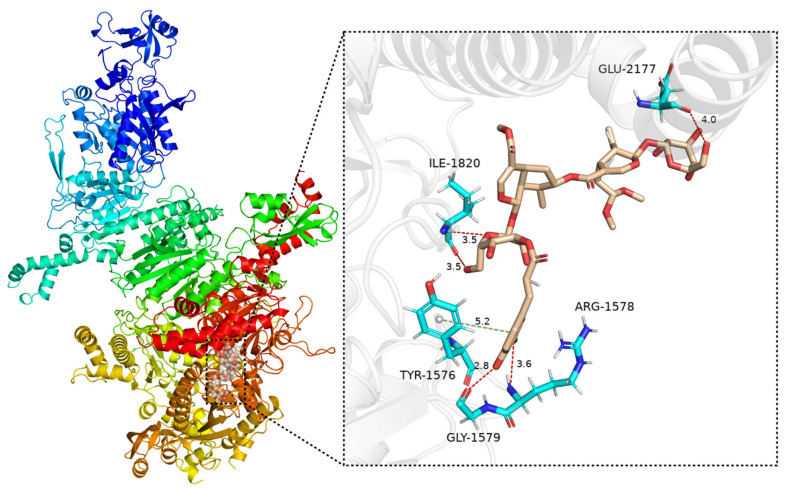
Molecular docking model of compound **2** bound to ACC1 (PDB ID: 3TVU), with various interactions shown as dotted lines (hydrogen bond: red; π-π: grayish green).

**Table 1 molecules-29-05980-t001:** ^1^H (400 MHz) and ^13^C (100 MHz) NMR data *^a^* (*δ* in ppm) of compounds **1** and **2** in CD_3_OD.

No.	1		2	
	*δ*_H_ (*J* in Hz)	*δ* _C_	*δ*_H_ (*J* in Hz)	*δ* _C_
1	5.52, d (5.5)	97.8	5.52, d (5.5)	97.8
3	7.43, br s	153.3	7.45, br s	153.3
4		112.0		112.0
5	2.89, m	29.5	2.91, m	29.4
6	2.06, m; 1.62, m	33.2	2.06, m; 1.62, m	33.2
7	4.50, dd (6.9, 4.4)	104.1	4.50, dd (7.3, 4.7)	104.1
8	5.74, m	135.9	5.74, m	135.9
9	2.67, m	45.4	2.70, m	45.4
10	5.26, br d (17.0)	119.8	5.31, br d (17.1)	119.8
	5.20, br d (10.6)		5.27, br d (10.2)	
11		168.3		168.3
1′	5.47, d (1.8)	95.5	5.34, d (2.0)	97.4
3′	7.23, br s	150.7	7.43, br s	152.5
4′		114.5		113.4
5′	2.93, m	30.9	3.11, m	32.4
6′	2.17, m; 1.77, m	39.6	2.31, m; 1.75, m	40.3
7′	5.10, m	78.6	5.20, m	78.3
8′	1.92, m	40.0	2.12, m	41.0
9′	2.22, m	47.2	2.12, m	47.1
10′	1.03, d (7.0)	12.8	1.08, d (5.6)	13.7
11′		168.7		169.3
1″	4.69, d (7.9)	100.0	4.68, d (8.0)	100.1
2″	3.19, m	74.6	3.19, m	74.6
3″	3.38, m	78.0	3.38, m	78.0
4″	3.28, m	71.5	3.28, m	71.5
5″	3.30, m	78.4	3.30, m	78.4
6″	3.93, dd (12.1, 1.5)	62.7	3.89, dd (12.0, 2.0)	62.7
	3.69, dd (12.1, 5.8)		3.70, dd (12.0, 5.9)	
1‴	4.86, d (8.1)	97.6	4.79, d (8.0)	100.1
2‴	4.79, dd (9.0, 8.1)	74.5	3.39, m	73.2
3‴	3.62, m	75.8	5.06, dd (9.3, 9.2)	78.8
4‴	3.38, m	71.7	3.52, dd (9.4, 9.3)	69.8
5‴	3.38, m	78.5	3.27, m	78.2
6‴	3.92, br d (12.4)	62.8	3.92, dd (12.0, 2.0)	62.5
	3.68, dd (overlapped)		3.66, dd (12.0, 6.3)	
1″″		127.6		127.7
2″″	7.03, s	115.3	7.04, s	115.1
3″″/4″″		149.8/147.0		149.8/147.0
5″″	6.78, d (8.0)	116.5	6.76, d (8.0)	116.5
6″″	6.94, d (8.0)	123.2	6.94, d (8.0)	123.0
7″″	7.47, d (15.9)	147.0	7.59, d (15.9)	146.9
8″″	6.16, d (15.9)	114.7	6.32, d (15.9)	115.3
9″″		168.0		169.0
7-OMe	3.28, s (6H)	53.5 or 52.8	3.29, s (6H)	53.6 or 52.7
11′-OMe	3.15, s	51.7	3.68, s	51.7

*^a^* Assignments were made by a combination of 1D and 2D NMR experiments.

**Table 2 molecules-29-05980-t002:** ^1^H (400 MHz) and ^13^C (100 MHz) NMR data *^a^* (*δ* in ppm) of compounds **3** and **4** in CD_3_OD.

No.	3		4	
	*δ*_H_ (*J* in Hz)	*δ* _C_	*δ*_H_ (*J* in Hz)	*δ* _C_
2		130.2		130.0
3	4.51, br d (11.3)	53.2	4.84, dd (3.7, 3.7)	52.7
5	3.89, dd, (11.7, 4.0)	59.6	4.12, dd (9.1, 5.7)	55.0
6	3.44, m; 3.01, m	24.0	3.39, m; 3.13, m	24.1
7/8		108.4/138.5		107.3/138.4
9	7.48, d (7.4)	119.8	7.47, d (7.8)	119.2
10	7.04, dd (7.4, 7.0)	120.7	7.04, dd (8.0, 7.2)	120.5
11	7.13, dd (8.0, 7.0)	123.5	7.14, dd (7.8, 7.2)	123.5
12	7.31, d (8.0)	112.2	7.36, d (8.0)	112.4
13		127.5		127.4
14	2.45, m; 2.25, m	34.6	2.36, m; 2.17, m	35.3
15	3.09, m	32.7	3.16, m	31.5
16		108.9		110.8
17	7.87, br s	157.3	7.64, br s	154.9
18	5.40, br d (17.8); 5.26, br d (10.4)	119.2	5.34, br d (17.2); 5.30, br d (10.3)	120.8
19	5.85, m	135.2	5.93, m	135.0
20	2.81, m	45.3	2.18, m	45.6
21	5.92, d (8.3)	97.6	5.63, d (7.1)	97.8
22		171.2		169.3
23		173.5		174.0
1′	5.27, d (5.5)	97.5	5.28, d (4.9)	97.6
3′	7.41, br s	152.7	7.44, br s	152.7
4′		113.0		113.1
5′	3.12, m	32.8	3.11, m	32.8
6′	2.35, m; 1.75, m	40.5	2.34, m; 1.76, m	40.4
7′	5.29, m	79.8	5.28, m	79.1
8′	2.16, m	41.3	2.14, m	41.1
9′	2.08, m	47.0	2.07, m	47.1
10′	1.09, d (6.7)	14.1	1.04, d (6.7)	13.9
11′		169.3		169.4
1″	4.84, d (7.7)	100.5	4.75, d (7.8)	100.4
2″	3.24, m	74.7	3.21, m	74.7
3″	3.40, m	78.0	3.38, m	78.0
4″	3.28, m	71.7	3.25, m	71.7
5″	3.41, m	78.8	3.25, m	78.6
6″	4.02, br d (11.7); 3.66, m	63.1	3.97, dd (12.0, 2.0); 3.65, m	62.8
1‴	4.65, d (7.7)	100.2	4.65, d (7.7)	100.2
2‴	3.17, m	74.6	3.17, m	74.7
3‴	3.35, m	77.9	3.38, m	78.0
4‴	3.23, m	71.6	3.25, m	71.6
5‴	3.29, m	78.4	3.34, m	78.4
6‴	3.88, br d (11.5); 3.61, m	62.8	3.89, dd (12.0, 2.0); 3.62, m	62.8
OMe	3.68, s	51.8	3.69, s	51.8

*^a^* Assignments were made by a combination of 1D and 2D NMR experiments.

**Table 3 molecules-29-05980-t003:** ^1^H (400 MHz) and ^13^C (100 MHz) NMR data *^a^* (*δ* in ppm) of compounds **5** and **6** in CD_3_OD.

No.	5		6	
	*δ*_H_ (*J* in Hz)	*δ* _C_	*δ*_H_ (*J* in Hz)	*δ* _C_
1*α*	1.13, dd (overlapped)	47.4	1.13, dd (overlapped)	47.3
1β	2.27, dd (12.4, 5.3)		2.27, dd (12.7, 5.3)	
2	3.54, m	74.2	3.54, m	74.1
3	3.76, d (9.1)	79.6	3.77, d (9.0)	79.6
4		151.5		151.5
5	1.75, br d (overlapped)	51.1	1.77, br d (12.6)	51.1
6	1.53, m	22.2	1.55, m	22.2
7	1.42, m	32.0	1.43, m	32.0
8		43.5		43.2
9	2.15, d (2.9)	53.4	2.17, d (2.8)	53.4
10		39.2		39.3
11	6.56, dd (11.8, 2.9)	126.5	6.50, dd (10.6, 2.8)	126.7
12	5.72, d (11.8)	127.6	5.77, d (10.6)	127.6
13		138.9		136.3
14		42.0		42.5
15	1.97, m; 1.07, m	26.1	1.07, m	26.3
16	2.23, m; 1.35, m	34.2	1.90, m; 1.68, m	33.4
17		47.3		47.7
18		136.8		134.9
19	2.88, m	38.5	3.22, m	37.5
20	1.70, m	36.2		137.6
21	1.96, m; 1.19, m	26.1	5.43, br d (5.0)	121.0
22	1.73, m	35.3	2.77, dd (16.0, 6.2)	38.9
	1.73, m		1.74, br d (16.0)	
23	5.16, 4.73, br s	105.0	5.17, 4.74, br s	105.5
25	0.75, s	16.6	0.76, s	16.6
26	0.90, s	17.1	0.93, s	17.3
27	1.00, s	20.9	0.99, s	20.2
28		180.3		182.4
29	1.05, d (7.2)	21.9	1.18, d (7.4)	21.4
30	0.92, d (7.1)	20.2	1.66, s	21.8

*^a^* Assignments were made by a combination of 1D and 2D NMR experiments.

**Table 4 molecules-29-05980-t004:** ACC1 and ACL inhibitory effects of compounds **2**, **8**, **17**, **29**, and **31**.

Compound	IC_50_ *^a^*	
	ACC1	ACL
**2**	9.6 ± 1.0 μM	>20
**8**	>20	3.6 ± 0.9 μM
**17**	10.3 ± 0.8 μM	2.0 ± 0.5 μM
**29**	>20	1.6 ± 0.4 μM
**31**	>20	4.7 ± 1.3 μM
ND 630 *^b^*	2.0 ± 0.1 nM	NT *^d^*
BMS 303141 *^c^*	NT *^d^*	0.3 ± 0.1 μM

*^a^* These data are expressed as the mean ± SEM of triplicated experiments. *^b^* The positive control for the acetyl CoA carboxylase 1 (ACC1) assay. *^c^* The positive control for the ATP-citrate lyase (ACL) assay. *^d^* NT: not tested.

## Data Availability

The data presented in this study are available in the article and Appendix A.

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
