# Peer review of "Bis-Iridoid Glycosides and Triterpenoids from Kolkwitzia amabilis and Their Potential as Inhibitors of ACC1 and ACL"

_molecules, 2024, doi:10.3390/molecules29245980_

Round 1

Reviewer 1 Report

Comments and Suggestions for Authors

Author Response

Comments 1: Introduction: the existing intricacies in clarifying the taxonomic relationship between the genera Kolkwitzia and Lonicera should be mentioned, since this chemotaxonomic perspective (Lines 29-30) was involved in the Abstract.

Response 1: Thank you for your feedback. Since 2013 we have initiated a dedicated research program aimed at systematically identify bioactive novel natural products from rare and endangered plants (REPs) in China. To date, we have investigated a dozen of REPs, resulting in the publication of 30+ related manuscripts as part of the series “Phytochemical and biological studies on rare and endangered plants endemic to China” (Parts I to XXXVIII). Consequently, in the Introduction section of this manuscript, we mainly primarily focus on the plant's chemical constituents and their biological activities. As suggested, the sentence “Previous studies have conducted a systematic and comprehensive investigations on the morphological phenotype traits and molecular phylogenetic [11,12].” has been incorporated into the Introduction section.

Comments 2: In general, the essence of chemotaxonomic significance of the compounds detected in this work should be underlined more clearly. Secoiridoid/iridoid-subtype dimers were previously identified within the Caprifoliaceae family, was it also done for Lonicera genus representatives? If not, why the findings of the manuscript (related to Kolkwitzia) are in support to the close phylogenetic relationship between Kolkwitzia and Lonicera (Line 327)?

Response 2: We sincerely thank the reviewer for the insightful suggestion. Secoiridoid/iridoid-subtype dimers have previously been identified within the Caprifoliaceae family (e.g. Phytochem. Lett. 2022, 49, 202−210; Phytochem. Lett. 2019, 33, 17−21; J. Nat. Prod. 2000, 63, 998−999). However, such identification has not been conducted for representatives of the Lonicera genus. Literature research reveals that plant taxonomy has traditionally been based on morphological phenotype traits but has progressively incorporated advanced techniques, such as molecular phylogenetics and chemotaxonomy analyses (Biol. J. Linn. Soc. 2005, 85, 407–415; Syst. Biol. 2005, 54, 844–851; Annu. Rev. Entomol. 2010, 55, 421–438; China Diversity 2022, 14, 289). In this study, chemotaxonomy analyses combined with previously reported morphological phenotype traits and molecular phylogenetic analyses on K. amabilis further substantiate the close phylogenetic relationship between Kolkwitzia and Lonicera (Mitochondrial DNA A 2017, 28, 296‒297; Sci. Hortic. 2014, 165, 190‒195).

Comments 3: Problem of ACC1 or ACL inhibitory activity within the iridoid glycosides and triterpenoids (of plant origin) implemented in treating metabolic diseases should be covered. The reader could remember that it is not a rare finding exclusively shown for chemical constituents from Caprifoliaceae species, but ordinary phenomenon. The matter is that iridoids and their glycosides are found in many families of plants, such as Cornaceae, Rubiaceae, Garryaceae, Eucommiaceae, Garryaceae, Ericaceae, Buddlejaceae, Globulariaceae, Hippuridaceae, Orobanchaceae, Plantaginaceae, Scrophulariaceae, Verbenaceae, Bignoniaceae, Lentibulariaceae, Martyniaceae, Myoporaceae, Lamiaceae, Apocynaceae, Altingiaceae, Davidiaceae, Monotropaceae, Pyrolaceae, Stylidiaceae, and Globulariaceae [Hussain et al., 2019].

Response 3: We thank the reviewer for their thoughtful observation. Iridoid glycosides and triterpenoids are widely distributed throughout the plant kingdom, and several of these compounds have demonstrated inhibitory effects on ACC1 or ACL. In the present study, K. amabilis, a member of the Caprifoliaceae family, was investigated. Consequently, we focused on iridoid glycosides and triterpenoids reported within the Caprifoliaceae family that exhibit ACC1 or ACL inhibitory activities. This finding is consistent with previous observations.

Comments 4: Compound 35 claimed as "vanillic" (likely "vanillic acid") (Line 105) does not coincide to its molecular structure in Figure S1 ("Phenylpropanoids"). Vanillic acid is 4-hydroxy-3- methoxybenzoic acid. Please correct a chemical name, or draw this structure not shared with phenylpropanoids 3234 and 36. Correspondingly, the text portion about "eight phenylpropanoids (32‒39, Figure S1)" from different parts of manuscript should be reconsidered depending on the actual structure of 35.

Response 4: We greatly appreciate the valuable suggestions! The molecular structure in Figure S1 is correct. The chemical name of compound 35 and its corresponding reference [45] have been correctly revised in the resubmitted manuscript.

Comments 5: It seems reasonable to divide the section "Results and Discussion" into subsections with subtitles, following "chemistry then biology" principle. In the present version of manuscript, the discussion on biological properties starts (Line 255) after eight pages of pure organic chemistry without transition. Molecular docking studies, and Chemotaxonomy, could also constitute separate subsections. Other proposed corrections to manuscript.

Response 5: We thank the reviewer for kindly pointing this out! We agree with this comment. In the revised version, the "Results and Discussion" section has been divided into four subsections: 2.1 Structure identification of compounds 16; 2.2 Anti-ACC1 and anti-ACL bioactivities of the isolated compounds; 2.3 Molecular docking simulation of compounds 2, 17, and 29; and 2.4 Chemotaxonomic significance.

Comments 6: Lines 15, 79: "include 13 iridoid glycosides (1‒4 and 7‒13)" should be replaced by "include 11 iridoid glycosides (1‒4 and 7‒13)".

Response 6: We thank the reviewer for bringing this to our attention. This error has been corrected in the revised manuscript.

Comments 7: Line 256: "malony-CoA" should be replaced by "malonyl-CoA".

Response 7: We greatly appreciate the valuable suggestions! "malony-CoA" has been replaced with "malonyl-CoA" in the revised manuscript.

Comments 8: Line 262: "NASH" should be spelled out, e.g. non-alcoholic steatohepatitis (NASH), or merely used without abbreviation.

Response 8: Thank you for your suggestion. "NASH" has been revised to "non-alcoholic steatohepatitis (NASH)" in the revised manuscript.

Reviewer 2 Report

Comments and Suggestions for Authors

This research manuscript shows the isolation and identification of iridoids and triterpenoids from K. amabilis. The isolation is accurately described, and the structure elucidation appears to be reasonable. In addition, some compounds inhibited enzymes associated with metabolic disorders. The manuscript is valuable and interesting. It fits well to this journal and its readership. I recommend acceptance after minor revision: 

Introduction, line 53: Please explain the meaning of ´´Weishi´´ for the non-Chinese readers.

Introduction, line 67: Please mention briefly some relevant diseases associated with ACC1 and ACL here. Please write K. amabilis in italics.

Results: Please explain why CD3OD was used as (polar) NMR solvent.

Line 182: Please correct ´´(Figure 4)correlations´´.

Table 4, Figure 5: Please specify the abbreviations of the tested enzymes in the footnote of the table and the caption of the figure.

Molecular docking: Please compare the binding modes and energies of compounds 2, 17, and 29 with the applied positive controls ND-630 and BMS-303141.

Author Response

Comments 1: Introduction, line 53: Please explain the meaning of “Weishi” for the non-Chinese readers.

Response 1: Yes, “Weishi” is spelling of the Chinese name.

Comments 2: Introduction, line 67: Please mention briefly some relevant diseases associated with ACC1 and ACL here. Please write K. amabilis in italics.

Response 2: Thank you for this valuable suggestion! The relevant diseases associated with ACC1 and ACL have been mentioned in Introduction. “K. amabilis” has been italicized.

Comments 3: Results: Please explain why CD3OD was used as (polar) NMR solvent.

Response 3: Yes, the isolated compounds are readily soluble in CD3OD.

Comments 4: Line 182: Please correct “(Figure 4)correlations”.

Response 4: Thanks. The phrase “(Figure 4)correlations” has been revised to “(Figure 4) correlations” in the revised manuscript.

Comments 5: Table 4, Figure 5: Please specify the abbreviations of the tested enzymes in the footnote of the table and the caption of the figure.

Response 5: Thanks! The full names of the ACC1 and ACL enzymes are provided in the footnote of table 4 and the caption of figure 5.

Comments 6: Molecular docking: Please compare the binding modes and energies of compounds 2, 17, and 29 with the applied positive controls ND-630 and BMS-303141.

Response 6: We greatly appreciate the reviewer’s valuable suggestions. In our previously studies, molecular docking studies on the positive controls ND-630 (for ACC1) and BMS-303141 (for ACL) were conducted (Bioorg. Chem. 2022, 120, 105630; J. Nat. Prod. 2023, 86, 1487‒1499). The active compounds (2, 17, and 29) in this study demonstrated distinct binding interactions compared to the positive controls.

Reviewer 3 Report

Comments and Suggestions for Authors

The paper reports an investigation of compounds isolated from Kolkwitzia amabilis. The compounds were characterized through spectroscopic data and electronic circular dichroism analyses. Their inhibitory activity against acetyl CoA carboxylase 1 (ACC1) and ATP-citrate lyase (ACL) were determined. The ligand-enzyme interaction of some of the compounds estimated as being active against these enzymes, were investigated with docking.

The following technical points need to be addressed:

Abstract:

Line 15 - “These compounds include 13 iridoid glycosides (1‒4 and 7‒13)” – the compounds are 11.

Line 22 – “Compound 2 exhibited a significant inhibitory effect against acetyl CoA carboxylase 1 (ACC1)” – according to the text line 267-268 “compounds 2 (IC50: 9.6 μM) and 17 (IC50: 10.3 μM) displayed moderate inhibitory effects against ACC1.”

2. Results and Discussion:

Line 79 – “These constituents include 13 iridoid glycosides (1‒4 and 7‒13, Figure 1)…” – the compounds are 11, not 13.

Line 105 – “vanillic (35)” – if the authors meant vanillic acid, compound 35 of Figure S1 is not vanillic acid.

Line 106 – “coniferin (36) [46]” – the reference needs to be confirmed as coniferin seems not to be provided in [46].

Lines 175-176, 189, 193 – “dipsaperine, which was previously isolated from the roots of Dipsacus asper [48]” – the reference needs to be confirmed as dipsaperine seems not to be provided in it. Dipsaperine is described in reference [51].

Lines 237-238 – reference [51] also needs to be confirmed for the listed compound.

Line 260 – “Firsocotat” – the name of the compound is Firsocostat.

Line 274 – “These data are expressed as the mean SEM of triplicated experiments” – it should be written “These data are expressed as the mean ± SEM of triplicated experiments”

Lines 290-298 – Some of the H-bond and salt bridge interactions are longer than the generally accepted maximal distance for these interactions (not longer than 4 Å), therefore the authors need to confirm these bonds and distances.

Materials and methods:

Line 467 - lit “{[75]: [α]D21 +44.0 (c 0.1, H2O)}” – in reference [75] it is written “ [α]D23 +44.0 (c 0.1, H2O)”

Line 475 – “following previously described procedures [9,15].” – The authors need to describe the procedures also in this paper as the reader may not have access to the referenced papers.

Line 480 – “ACC1 (PDB ID: 3TVU) and ACL (PDB ID: 6HXH)” - the references for the pdb codes need to be provided.

The versions of the software used need to be provided.

A citation for AutoDock tools to be provided.

Line 487 – “The number of GA runs” – the abbreviation GA needs to be explained.

Supplementary materials:

Figures S5 and S6 – in Figures 6 and S6 the bond distances are presented, they need to be presented also in Figure S5 for consistency.

Author Response

Comments 1: Line 15-“These compounds include 13 iridoid glycosides (1‒4 and 7‒13)”–the compounds are 11.

Response 1: Many thanks to the reviewer for the valuable suggestions. This error has been corrected in the revised manuscript.

Comments 2: Line 22–“Compound 2 exhibited a significant inhibitory effect against acetyl CoA carboxylase 1 (ACC1)” – according to the text line 267-268 “compounds 2 (IC50: 9.6 μM) and 17 (IC50: 10.3 μM) displayed moderate inhibitory effects against ACC1.”

Response 2: We thank the reviewer for kindly pointing this out! The sentence “Compound 2 exhibited a significant inhibitory effect against acetyl CoA carboxylase 1 (ACC1)” has been revised to “Compound 2 exhibited a moderate inhibitory effect against acetyl CoA carboxylase 1 (ACC1)”.

Comments 3: Line 79 – “These constituents include 13 iridoid glycosides (1‒4 and 7‒13, Figure 1)…” – the compounds are 11, not 13.

Response 3: Thanks. This error has been corrected in the revised manuscript.

Comments 4: Line 105 – “vanillic (35)” – if the authors meant vanillic acid, compound 35 of Figure S1 is not vanillic acid.

Response 4: Many thanks for the valuable suggestions! The molecular structure in Figure S1 is correct. The chemical name of compound 35 and the corresponding reference [45] have been corrected accordingly in the resubmitted manuscript.

Comments 5: Line 106 – “coniferin (36) [46]” – the reference needs to be confirmed as coniferin seems not to be provided in [46].

Response 5: Many thanks for the valuable suggestions! The reference [46] for compound 36 has been corrected accordingly in the revised manuscript.

Comments 6: Lines 175-176, 189, 193 –“dipsaperine, which was previously isolated from the roots of Dipsacus asper [48]”–the reference needs to be confirmed as dipsaperine seems not to be provided in it. Dipsaperine is described in reference [51]. Lines 237-238 – reference [51] also needs to be confirmed for the listed compound.

Response 6: We thank the reviewer for kindly pointing this out. References [48] and [51] have been correctly cited in the revised manuscript.

Comments 7: Line 260 – “Firsocotat” – the name of the compound is Firsocostat.

Response 7: Thanks. “Firsocotat” has been replaced by “Firsocostat”.

Comments 8: Line 274 – “These data are expressed as the mean SEM of triplicated experiments” – it should be written “These data are expressed as the mean ± SEM of triplicated experiments”

Response 8: Many thanks! The sentence “These data are expressed as the mean SEM of triplicated experiments” has been revised to “These data are expressed as the mean ± SEM of triplicated experiments”.

Comments 9: Lines 290-298 – Some of the H-bond and salt bridge interactions are longer than the generally accepted maximal distance for these interactions (not longer than 4 Å), therefore the authors need to confirm these bonds and distances.

Response 9: Many thanks for pointing this out! We agree with this comment and confirm that all bonds (including hydrogen bonds, salt bridge, and π-π stacking) and their corresponding distances are correct in this article. Bonds longer than 4 Å may result from interactions between multiple active sites in the active compound and the ACC1/ACL protein.

Comments 10: Line 467-lit “{[75]: [α]D 21 +44.0 (c 0.1, H2O)}”–in reference [75] it is written “[α]D 23 +44.0 (c 0.1, H2O)”

Response 10: We thank the reviewer for kindly pointing this out. “{[75]: [α]D 21 +44.0 (c 0.1, H2O)}” has been revised to “[{[75]: [α]D 23 +44.0 (c 0.1, H2O)} ” in the revised manuscript.

Comments 11: Line 475 – “following previously described procedures [9,15].” – The authors need to describe the procedures also in this paper as the reader may not have access to the referenced papers.

Response 11: Many thanks for the valuable suggestions! We agree with this comment. The testing procedures have been included in the revised version.

Comments 12: Line 480 – “ACC1 (PDB ID: 3TVU) and ACL (PDB ID: 6HXH)” - the references for the pdb codes need to be provided.

Response 12: We thank for the valuable suggestions! The references for the PDB codes of ACC1 and ACL have been provided.

Comments 13: The versions of the software used need to be provided. A citation for AutoDock tools to be provided.

Response 13: Many thanks for the valuable suggestions! The software version and citation for AutoDockTools have been provided in the revised manuscript.

Comments 14: Line 487 – “The number of GA runs” – the abbreviation GA needs to be explained.

Response 14: We thank for the valuable suggestions! “The number of GA runs” has been revised to “The number of GA (Genetic Algorithm) runs” in the revised manuscript.

Comments 15: Figures S5 and S6 – in Figures 6 and S6 the bond distances are presented, they need to be presented also in Figure S5 for consistency.

Response 15: Thanks for this valuable suggestion! The bond distances have been included in the revised Figure S5.